# Serine 31 Phosphorylation-Driven Regulation of AGPase Activity: Potential Implications for Enhanced Starch Yields in Crops

**DOI:** 10.3390/ijms242015283

**Published:** 2023-10-18

**Authors:** Guowu Yu, Yuewei Mou, Noman Shoaib, Xuewu He, Lun Liu, Runze Di, Nishbah Mughal, Na Zhang, Yubi Huang

**Affiliations:** 1State Key Laboratory of Crop Gene Exploration and Utilization in Southwest China, Sichuan Agricultural University, Chengdu 611130, China; mouyuewei@stu.sicau.edu.cn (Y.M.); noman@cib.ac.cn (N.S.); liulun@stu.sicau.edu.cn (L.L.); dirunze@stu.sicau.edu.cn (R.D.); 2020601004@stu.sicau.edu.cn (N.M.); 10024@sicau.edu.cn (Y.H.); 2National Demonstration Center for Experimental Crop Science Education, College of Agronomy, Sichuan Agricultural University, Chengdu 611130, China; 3College of Life Science, Sichuan Agricultural University, Ya’an 625014, China; hxiaowu@stu.sicau.edu.cn; 4College of Science, Sichuan Agricultural University, Chengdu 611130, China; 72008@sicau.edu.cn

**Keywords:** AGPase, shrunken2, phosphorylation, enzyme activity, subcellular localization

## Abstract

ADP-Glc pyrophosphorylase (AGPase), which catalyzes the transformation of ATP and glucose-1-phosphate (Glc-1-P) into adenosine diphosphate glucose (ADP-Glc), acts as a rate-limiting enzyme in crop starch biosynthesis. Prior research has hinted at the regulation of AGPase by phosphorylation in maize. However, the identification and functional implications of these sites remain to be elucidated. In this study, we identified the phosphorylation site (serine at the 31st position of the linear amino acid sequence) of the AGPase large subunit (Sh2) using iTRAQ^TM^. Subsequently, to ascertain the impact of Sh2 phosphorylation on AGPase, we carried out site-directed mutations creating Sh2-S31A (serine residue replaced with alanine) to mimic dephosphorylation and Sh2-S31D (serine residue replaced with aspartic acid) or Sh2-S31E (serine residue replaced with glutamic acid) to mimic phosphorylation. Preliminary investigations were performed to determine Sh2 subcellular localization, its interaction with Bt2, and the resultant AGPase enzymatic activity. Our findings indicate that phosphorylation exerts no impact on the stability or localization of Sh2. Furthermore, none of these mutations at the S31 site of Sh2 seem to affect its interaction with Bt2 (smaller subunit). Intriguingly, all S31 mutations in Sh2 appear to enhance AGPase activity when co-transfected with Bt2, with Sh2-S31E demonstrating a substantial five-fold increase in AGPase activity compared to Sh2. These novel insights lay a foundational groundwork for targeted improvements in AGPase activity, thus potentially accelerating the production of ADP-Glc (the primary substrate for starch synthesis), promising implications for improved starch biosynthesis, and holding the potential to significantly impact agricultural practices.

## 1. Introduction

Starch, owing to its abundant availability and ease of modification, plays a crucial role as a primary raw material across multiple sectors, including food, medicine, ethanol production, biodegradable materials, and green water treatment agents [1,2,3]. Amid the ongoing scarcity of per capita resources and the escalating focus on environmental conservation in China, maize starch has emerged as a promising alternative, leveraging its high yield [4] and versatility in product processing and reprocessing [5,6]. Thus, it is increasingly critical to examine the synthesis of maize starch and devise strategies to augment starch synthesis. The intricate process of starch biosynthesis primarily involves catalytic enzymes including, adenosine diphosphate glucose pyrophosphorylase (AGPase), starch synthase (SS), starch branching enzyme (SBE), and starch debranching enzyme (DBE). AGPase is a crucial rate-limiting enzyme in starch biosynthesis and is primarily responsible for catalyzing glucose-1-phosphate Glc-1-P and ATP to form adenosine diphosphate glucose (ADP-Glc), which is the key substrate for starch synthesis [7].

In higher plants, AGPase is a hetero-tetramer constituted of two small and two large subunits (α2β2). Although the proteins encoded by these subunits are similar, notable differences exist. For instance, the small subunit exhibits a higher level of conservation (85–95%) compared to the large subunit (50–60%), and the molecular weight of the large subunit (54–60 kDa) is slightly superior to that of the small subunit (51–55 kDa) [8]. Optimal AGPase activity is contingent on the interaction between these two subunits [9]. While the large subunit serves as the regulatory subunit and modulates its metabolic activity, the small subunit operates as the catalytic subunit and executes its metabolic function [10]. This segregation of roles within the hetero-tetrameric structure of AGPase underscores its intricate function in starch biosynthesis.

In numerous species, AGPase predominantly resides in plastids, with plastid-based AGPase being the exclusive source of ADP-Glc which is the precursor in starch synthesis [11]. Moreover, its expression patterns underscore its significant role in starch biosynthesis. There was increased Sh2 expression in 0–20 DAP endosperm, followed by a decrease at the mRNA level between 20–40 DAP [12]. Yet, in the endosperm of grains, ADP-Glc primarily stems from the cytoplasmic AGPase [13]. Thus, AGPase is situated both in the cytoplasm and plastid of grain endosperm with a considerable proportion (85–95%) of its activity occurring in the cytoplasm [14]. These observations indicate the presence of a distinct pathway in cereal endosperm starch synthesis, necessitating an ADPG transporter for the relocation of cytoplasmic AGPase-generated ADP-Glc into the plastid [15,16]. In higher plants, AGPase encompasses numerous isozyme forms, with size subunits encoded by discrete genes and tissue-specific AGPase expression [17]. In maize, three primary genes encode the large subunit [18,19,20]. Among these, Sh2 encodes the large subunit of the cytoplasmic gene, while *AGPLEMZM* and *AGPLLZM* encode the large subunit in the embryo and leaf tissue, respectively [21]. The size of the subunit is contingent on the localization of AGPase, resulting in differential activities of AGPase at distinct cellular locations [22,23].

The regulation of AGPase varying across tissues and cells, impacts its activity. Its activity is synergistically modulated through various regulatory means, including redox [24] allosteric [25], and transcriptional approaches [26,27]. With advancements in experimental techniques, including mass spectrometry and affinity chromatography, our understanding of proteomes and phosphoproteomes has grown. Prior studies have identified phosphorylated AGPase across diverse species and tissues [28,29]. Research on grains has reported AGPase subunit phosphorylation. For instance, differential gel electrophoresis analysis of germinated wheat seeds suggested that the small subunit of plastid AGPase is phosphorylated, although the specific phosphorylation site remained undetermined [30]. Previously in our laboratory, alkaline phosphatase treatment of samples for gel activity measurement led to lower enzymatic activity compared to the phosphorylated control, implying a relationship between AGPase activity and phosphorylation [31]. Recently, studies have confirmed the phosphorylation of AGPase large subunit in wheat endosperm [32], with in vitro experiments using recombinant AGPase from wheat endosperm and calcium-dependent protein kinase demonstrating that phosphorylation primarily occurs in the large subunit.

In this study, a specific polyclonal antibody against Sh2 was generated via the immunization of New Zealand white rabbits. Endosperm proteins of maize at 15, 20, and 27 days after pollination (DAP) were enriched using Phos-tag™ agarose. Sh2 was identified through Western blotting; however, binding to Phos-tag™ was lost when samples underwent alkaline phosphatase (ALP) treatment, indicating Sh2 phosphorylation in maize endosperm. Furthermore, the phosphorylation site of Sh2 at serine 31 (S31) was identified through iTRAQ^TM^ analysis. Employing site-directed mutagenesis, the serine residue was mutated to alanine (A) to simulate dephosphorylation and to aspartic acid (D) and glutamic acid (E) to mimic phosphorylation. We initiated a preliminary investigation of the effects of these phosphorylation changes on the subcellular localization of Sh2, its binding with Bt2, and its enzyme activity. Findings revealed that the S31A/D/E mutations did not alter the subcellular localization of Sh2 or impact its interaction with Bt2. Nevertheless, the mutations simulating phosphorylation enhanced AGPase activity. These results lead us to conclude that phosphorylation may also represent a mechanism for AGPase activity regulation during the process of starch biosynthesis.

## 2. Results

### 2.1. Preparation and Evaluation of Sh2 Antibody against AGPase

We generated specific antibodies against Sh2 by immunizing New Zealand white rabbits with purified GST-Sh2 protein as an antigen. Molecular cloning and vector construction facilitated the successful acquisition of the pGEX-6T-GST-Sh2 vector. This construct allowed the fusion protein between GST and the target protein to be severed by the TEV enzyme (Appendix A). The expression of GST-Sh2 protein was achieved in BL21 bacterial cells post-transfer of pGEX-6T-GST-Sh2, induced by 0.2 mm IPTG, and the protein was subsequently purified using the BBI Solutions GST protein purification kit (Appendix A). Following the established antibody production protocol, the immunization antiserum was collected and purified, allowing for subsequent target protein detection. To validate the specificity of the Sh2 antibody, the GST-Sh2 protein was cleaved with the TEV enzyme over various periods from 0 to 6 h. Western blot (WB) analysis demonstrated that the Sh2 antibody identified both Sh2 and GST (Appendix A). This confirmed the ability of the Sh2 antiserum to specifically recognize Sh2 protein in vitro.

### 2.2. Analysis of Immunoprecipitation and Sh2 Expression of Maize Endosperm Protein

The successfully prepared specific Sh2 antibody was then investigated for its ability to recognize Sh2 in maize endosperm. WB was performed using the Sh2 antibody on 20 DAP maize endosperm samples (Figure 1B). As a negative control, the pre-immunization rabbit serum was unable to identify the Sh2 protein (Figure 1A), confirming the successful preparation of the Sh2 antibody. Further, to demonstrate the in vivo specificity of the Sh2 antibody, we performed immunoprecipitation using the 27 DAP endosperm lysate. Subsequent WB of the Sh2 antibody immunoprecipitation revealed specific recognition and precipitation of the Sh2 protein, a characteristic not seen with the IgG control (Figure 1C). Further, we analyzed the Sh2 protein expression in the 5–35 DAP endosperm, correlating it with previous reports suggesting increased Sh2 expression in 0–20 DAP endosperm, followed by a decrease at the mRNA level between 20–40 DAP. Our Western blotting results, obtained with the Sh2 antibody, echoed this observation, revealing an increasing trend in Sh2 protein levels in the 0–30 DAP endosperm and then decreasing quickly at 35 DAP (Figure 1D). Collectively, these results validate the prepared Sh2 antibody as an effective tool for immunoblotting and IP detection.

### 2.3. Phos-Tag^TM^ Enrichment of Sh2 Phosphorylated Proteins

Maize endosperm proteins extracted from the developmental stages 15, 20, and 27 DAP were subjected to Phos-tag™ enrichment to isolate phosphorylated proteins. After the removal of non-specifically bound proteins via water washing, WB was carried out using an Sh2-specific antibody to identify Sh2 in the enriched phosphorylated protein sample. Even though Phos-tag™ does not perfectly enrich all phosphorylated proteins, clear bands corresponding to Sh2 proteins were evident. Upon treatment of identical maize endosperm protein samples with alkaline phosphatase, these Sh2 bands either disappeared or were significantly weakened (Figure 2A). To further demonstrate the phosphorylation of Sh2, we did the time course of alkaline phosphatase treatment. The results show phosphorylated Sh2 enriched with Phos-tag^TM^ beads decreased in the indicated treatment time (Figure 2B). These findings suggest that Sh2 likely undergoes phosphorylation during the process of starch biosynthesis.

### 2.4. iTRAQ^TM^ Identification of Sh2 Phosphorylation Sites

In a previous study, we utilized Phos-tag^TM^ technology to enrich phosphorylated proteins from endosperm samples collected at different stages of maize pollination and identified a large number of phosphorylated proteins involved in the starch synthesis pathway through mass spectrometry analysis [31]. Among these proteins, peptides corresponding to the AGPase were detected. During mass spectrometry analysis, numerous peptides were identified. Among them, a particular peptide that contains S31 was chosen for further investigation due to its increased likelihood of phosphorylation. Based on these findings, we speculated that Sh2 may be phosphorylated by kinases. Our results point towards the phosphorylation of Sh2 at S31 (Table 1; Figure 3). This finding elucidates the phosphorylation of Sh2 which may occur at S31 in the maize endosperm.

Subsequently, to further explore the functional impact of phosphorylation at S31 of Sh2, we employed site-directed mutagenesis using the QuickMutation™ Gene Site-directed Mutation Kit (Beyotime, Shanghai, China). The S31 of Sh2 in the puG-221-Sh2 plasmid was mutated to A, D, and E according to the manufacturer’s instructions. Following mutagenesis and transformation into DH5α recipient cells, positive colonies were identified and sent for sequencing to confirm the successful introduction of the mutations (Appendix A). These confirmed mutated clones were then used for amplification of the mutated Sh2, paving the way for future recombination with required vectors.

### 2.5. Interaction between Sh2-S31 Phosphorylation Site Mutation and Bt2

Using a combination of vector construction and yeast two-hybrid techniques, we generated yeast vectors PGBK-T7-Sh2, PGBK-T7-Sh2-S31A/D/E, PGAD-T7-Bt2, and PGBK-T7-Sh2-S31A, along with fluorescent bi-molecular vectors E2884-Sh2 (pSAT6-nEYFP-C1), E2884-Sh2-S31A/D/E, and E3108-Bt2 (pSAT6-cEYFP-C1). We employed yeast two-hybrid assays using the constructed yeast vectors and the resulting interactions were analyzed. Confocal microscopy revealed a distinct yellow fluorescence signal indicative of successful transformation. Notably, both the yeast two-hybrid and bimolecular fluorescence complementation experiments affirmed that the S31A/D/E mutations in Sh2 did not hinder its interaction with Bt2 (Figure 4A,B), suggesting that these mutations do not detrimentally affect the functionality of the protein complex.

### 2.6. Subcellular Localization of Mutated Sh2-S31

High-concentration plasmids (10 µg) of pCAMBIA2300-Sh2 and pCAMBIA2300 -Sh2-S31A/D/E were transformed into maize etiolated seedling protoplasts or maize endosperm protoplasts. Following 12–14 h of incubation under dark conditions, the distribution of the green fluorescence signal was observed using confocal microscopy. The results demonstrated that Sh2 acts as a cytoplasmic protein in the endosperm of monocot plants and that the subcellular localization of Sh2 remains unaffected by the mutations (Figure 5).

### 2.7. AGPase Activity Afterward Phosphorylation Site Mutation

In our previous investigation, we established phosphorylation as a novel regulatory mechanism governing AGPase activity, thereby affirming the impact of Sh2 phosphorylation on AGPase function. To generate the Pug221-Sh2-S31A/D/E mutants, we introduced specific mutations into the previously constructed puG221-SH2 expression vector. Subsequently, maize endosperm protoplasts were transformed with Pug221-sh2 and PUG221-Sh2-S31A/D/E vectors, along with PUG221-Bt2 and pUG221 control vectors, and subjected to a 16- to 18-hour period of dark expression. ATP levels were assessed using the Beyotime assay kit, while GUS activity was measured to evaluate the efficiency of transformation. Analysis reveals that among the A and D mutations, the Glu mutation at position SH2-31 exhibited the highest level of activity, surpassing that of the wild-type Sh2 protein (Figure 6).

## 3. Discussion

AGPase is the rate-limiting enzyme in starch biosynthesis and plays a crucial role in catalyzing the formation of ADP-Glc by utilizing Glc-1-P and ATP [7]. The activity of AGPase is closely associated with starch synthesis and its inhibition leads to impaired starch production [33]. AGPase activity is regulated differently in various tissues and cells, which influences its overall functionality [12,22,23]. To respond promptly to environmental changes, AGPase activity is modulated through multiple regulatory mechanisms, including redox regulation, allosteric regulation [25], and transcriptional control [26,27], which have been extensively studied. In this study, we successfully obtained the Sh2 antibody and investigated AGPase phosphorylation using Phos-tag^TM^ and iTRAQ^TM^ techniques. Our results revealed the phosphorylation of the Sh2-S31 subunit during starch accumulation in grains. Building upon our previous experiments on AGPase activation [31], we propose that phosphorylation of AGPase may serve as a significant regulatory mechanism affecting its activity. Phos-tag^TM^, a binuclear metal complex, selectively binds phosphate monoesters in aqueous solutions. We employed the first application of Phos-tag^TM^, utilizing Phos-tag^TM^ derivatives and hydrophilic chromatography to enrich phosphorylated Bt2 and Sh2 peptides [34]. Alkaline phosphatase, a phosphatase enzyme capable of dephosphorylating substrates, was used to remove phosphorylation from Sh2. As a result, the binding of Sh2 to agarose-Phos-tag^TM^ was significantly diminished or even eliminated. Similar findings were observed in previous studies on the Bt2 subunit of AGPase [30], further reinforcing the notion that dephosphorylated AGPase subunits fail to bind to agarose-Phos-tag^TM^, thus indicating the presence of phosphorylation. In conclusion, the data presented in this study broadened our understanding of the molecular interactions between Sh2 and Bt2 proteins during maize endosperm development. They also suggest that the S31A/D/E mutations do not disrupt the interaction or potential functionality of the Sh2-Bt2 protein complex.

Furthermore, iTRAQ^TM^ analysis of mass spectrometry data on Sh2 peptides from 25 DAP maize endosperm AGPase revealed the phosphorylation of Sh2-S31. Collectively, indirect and direct evidence suggests the phosphorylation of Sh2 in vivo. Over the past decade, phosphorylated proteomics studies have reported phosphorylation events in AGPase from diverse species and tissues [28,29]. Notably, several studies conducted on grains have documented phosphorylation of AGPase subunits. For instance, analysis of germinated wheat seeds using differential gel electrophoresis demonstrated phosphorylation of the small subunit of plasmid AGPase, although without identification of the exact phosphorylation site [30]. Additionally, investigations on maize endosperm have revealed the enrichment of phosphorylated proteins, particularly in the small subunit Bt2, indicating the presence of phosphorylation in protein extracts. Recent studies have further confirmed the phosphorylation of the large subunit of AGPase in wheat endosperm, demonstrating that enzyme phosphorylation increases with grain development and exhibits a positive correlation with AGPase activity and starch content in endosperm extracts [32].

Protein phosphorylation has been shown to alter protein conformation, influencing protein–protein interactions, and consequently affecting the function of these proteins [35,36]. These changes could impact the protein activity, its localization within the cell, and the stability of complexes it forms with other proteins [37]. An investigation into the phosphorylation of the barley stripe Mosaic virus gamma Ser-96-b protein found that phosphorylation at key sites was integral to the protein function [38]. It was observed that a non-phosphorylated mutant (S96A) of γ B had significantly reduced the viral suppressor of RNA silencing (VSR) activity, compared to the phosphorylation-simulated mutant (S96D) and the wild-type γ B, which had similar VSR activities. This research suggests that phosphorylation-induced charge changes can significantly modify protein function. In a related study, the phosphorylation status of the Tau protein was found to directly influence its localization within the cell [39]. The N-terminal region of the Tau protein, when phosphorylation was simulated by substituting serine/threonine residues with glutamate, was found to hinder its membrane localization in transfected cells [31]. Furthermore, it is known that the phosphorylation of enzymes like SBEI and SP (starch phosphorylase) in maize is crucial for the formation of active multi-protein complexes [40,41,42]. Additionally, studies on wheat endosperm found that phosphorylation is necessary for the formation of the SS-SBEII protein complex and the protein complex between SBEIIb and SP [42]. Research on the Sh2 gene in maize, which codes for the large subunit of AGPase, suggests that its phosphorylation may impact the activity, localization, and binding of AGPase [31]. Experiments using the yeast two-hybrid method and bimolecular fluorescence complementation have shown that certain mutations in the Sh2 gene (Sh2-S31A/D/E) do not affect its interaction with Bt2. Moreover, the Sh2-Ser31D/E mutants, which simulate phosphorylation, displayed increased activity potentially through a mechanism that involves interaction with other proteins or alteration of the enzyme conformation (which warrants further exploration), when transiently expressed in endosperm protoplasts suggesting its potential for a better target to enhance the starch content. Given these findings, future work should aim to validate these results in vivo and examine the mechanistic underpinnings of this observed effect. This could involve investigations into the structural changes induced by these mutations, the potential interaction partners of the mutated Sh2 subunit, and the broader impact of these mutations on starch biosynthesis and plant physiology. Such studies will undoubtedly enrich our understanding of the regulatory complexity of AGPase and provide novel avenues for starch improvement.

## 4. Materials and Methods

### 4.1. Plant Materials

In this study, the experimental material used was the maize inbred line Mo17. The seeds were obtained from Sichuan Agricultural University and planted at the Agriculture Research and Development Base of the Sichuan Agricultural University. To preserve the seeds for future use, corn seeds at different developmental stages (10 DAP (days after pollination), 15 DAP, 20 DAP, 25 DAP, 30 DAP, and 35 DAP) were rapidly frozen using liquid nitrogen and stored at −80 °C. For the isolation of endosperm protoplasts, the Mo17 maize plants were grown at the Agriculture Research and Development Base and used for the experiment. Etiolated maize seedlings of Mo17, which were utilized to separate leaf protoplasts, were grown in a constant temperature incubator set at 28 °C with a light period of 16 h and a darkness period of 8 h. To ensure sterility, the seeds were disinfected with 3% H_2_O_2_ for 5 min before being planted in vegetative soil. After germination, the seedlings were transferred to dark conditions for further cultivation. In the Phos-tag^TM^ enrichment assay, three independent biological replicates of maize endosperm samples were collected simultaneously and pooled together to create a proteomic analysis bank.

### 4.2. Protein Expression and Purification

To create the GST-gene fusion system protein expression vector PGEX-6T-1-SH2, the Sh2 gene was incorporated into the PGEX-6T-1 vector. The Sh2 homologous recombination primers used for this purpose were as follows: Sh2-F: AAAACCTGTATT TTCAGGGATCCATGCAGTTTGCACTTGCATTGGACACG, Sh2-R: CAGTCACGATG CGGCCGCTCGAGCTATATGACAGACCCATCGTTGATGGTTG. To express and purify the Sh2 protein, the procedures outlined in the GST gene fusion system handbook provided by GE Healthcare (Piscataway, NY, USA) were followed. Briefly, the *E. coli* BL21 transformation cells carrying the PGEX-6T-1-SH2 construct were cultured in 0.1 L of LB (Luria–Bertani) medium supplemented with 50 µg/mL of ampicillin. The culture was incubated at 37 °C with continuous shaking at 150 g. Afterward, 0.5 mM IPTG was added to induce protein expression, and the incubation temperature was lowered to 28 °C for 6 hours. Samples were collected every 2 h by centrifuging bacterial cells at 4200× *g* for 10 min at 4 °C. The bacterial cells were resuspended in a PBS buffer solution (137 mM NaCl, 10 mM Na2HPO4, 2.7 mM KCl, 1.8 mM KH2PO4, pH 7.4), and recombinant proteins were extracted using ultrasound. The extract was then centrifuged at 12,000× *g* at 4 °C for 5 min to separate soluble proteins. Subsequently, the proteins in the total extract (before centrifugation), soluble fraction (supernatant), and insoluble fraction (precipitation) were separated using SDS-PAGE and visualized using Coomassie bright blue (CBB) staining. The purification involved separating the eluted protein using SDS-PAGE and subsequently removing and recovering the target protein bands from the gel via electro-elution. Following protein concentration measurement with the Bradford reagent (catalog no. p0006, Beyotime), the recombinant protein served as an antigen for the development of a polyclonal antibody.

### 4.3. Rabbit Breeding, Anti-Serum Preparation, and Antibody Purification

Da Shuo Experimental Animal Company provided three-month-old New Zealand white rabbits (weighing 2 kg) for the study. The rabbits were housed in the animal core facility, following the approved procedures of the Animal Care and Use Committee of Sichuan Agricultural University (approval no. 20160320, Chengdu, China). After one week of acclimation, rabbits were immunized subcutaneously with 500 µg of purified recombinant GST-Sh2 fusion protein emulsified with 500 µL of Freund’s complete adjuvant at a ratio of 1:1 (*v*/*v*). Two weeks after the first immunization, the rabbits were boosted with five additional subcutaneous injections with 500 µg of the purified protein mixed with 500 µL of Freund’s incomplete adjuvant per injection at a ratio of 1:1 every week [43]. Venous blood samples were collected after three injections. The resulting antiserum containing the polyclonal maize antibody was applied to a column containing a mixture of 50% protein A and 50% protein G. The column was washed with ice-cold TBS (50 mM Tris-HCl, pH 7.4, 150 mM NaCl, and 0.05% sodium azide). The antiserum was then thawed in ice water and clarified through centrifugation at 15,000× *g* for 5 min at 4 °C. Approximately 3 mL of the clarified antiserum was added to the column, followed by washing with TBS buffer. Elution buffer with pH 2.7 and pH 1.9 (100 mM glycine pH 2.7 and 100 mM glycine pH 1.9, respectively), was gently introduced to the column, yielding approximately 0.4 mL fractions in the collection tubes. These fractions were neutralized using NB buffer (1 M Tris-HCl, pH 8.0; 1.5 M NaCl; 1 mM EDTA; 0.5% sodium azide) to adjust the pH to approximately 7.4. The resulting purified antibodies were utilized in immunoblot and immunoprecipitation experiments. Negative controls were established using pre-immune sera for each of the antibodies mentioned above. These controls demonstrated no cross-reaction with proteins from maize endosperm lysates or co-immunoprecipitation experiments.

### 4.4. Extraction and Determination of Plant Proteins

Proteins were isolated from grains using a phenol extraction method described by [44]. Briefly, 3 grams of frozen tissue powder was re-suspended in a cool extract buffer consisting of 100 mM EDTA, 50 mM Borax, 50 mM vitamin C, 1% PVPP (*w*/*v*), 1% Triton X-100 (*v*/*v*), 2% β-Mercaptoethanol (*v*/*v*), and 30% sucrose (*w*/*v*) in a total volume of 5 mL. The mixture was gently agitated at room temperature for 5 min, followed by the addition of two volumes of tris-saturated phenol (pH 8.0) and further agitation for 10 min. After centrifugation at 4 °C for 15 min at 15,000× *g*, the supernatant was carefully transferred to a new centrifuge tube. An equal volume of protein extraction buffer was added to the new tube, and the mixture was centrifuged under the same conditions after vortexing for 10 min. The resulting supernatant was transferred to a new centrifuge tube, and 5 times the volume of ammonium sulfate precipitation buffer was added. The tube was then placed at −20 °C for at least 6 h. Following centrifugation under the aforementioned conditions, the protein pellet was re-suspended and washed twice with cold methanol and acetone sequentially. After each wash, the protein was briefly centrifuged at 15,000× *g* for 5 min at 4 °C, and the supernatant was carefully removed. Finally, the washed protein was air-dried or re-suspended in lysis buffer (9 M Urea, 2% Chaps, 13 mM DTT, 1% IPG buffer) and stored at −80 °C. The protein concentration was determined using the Bio-Rad protein assay, with bovine serum albumin as the standard, following the manufacturer’s instructions. SDS-PAGE gel electrophoresis and Western blotting (WB) were performed using the isolated protein extract.

### 4.5. Phos-TagTM Enrichment, SDS-PAGE and Immunoblotting

To enrich phosphoproteins, Zn^2+^-Phos-tag^TM^ agarose was utilized following the instructions provided by the manufacturer, Wako Pure Chemical Industries Ltd., Hiroshima, Japan. In brief, a total maize endosperm cell lysate sample containing 200 µg of protein was mixed with 200 mm of Zn^2+^-Phos-tag^TM^ agarose. For the dephosphorylation assay, alkaline phosphatase was added to the cell lysate. The binding assay was conducted at 4 °C for 4 h, followed by three washes with a washing buffer (0.1 M Tris-CH_3_COOH, 1.0 M CH_3_COONa, pH 7.5). Elution buffer (0.1 M Tris-CH_3_COOH, 1.0 M NaCl, 10 mM NaH_2_PO_4_-NaOH, pH 7.5) was utilized to elute the phosphoproteins. The proteins bound to Zn^2+^ Phos-tag^TM^ agarose were separated by electrophoresis. Additionally, 20 µg of maize endosperm cell lysates were subjected to SDS-PAGE as a control. The SDS polyacrylamide gels consisted of a 10% acrylamide separation gel and a 5% stacking gel. The immunoblotting was performed as described in [45]. The resolved proteins were transferred to nitrocellulose membranes (GE Healthcare Life Sciences, Cat: 10600003) via electrophoretic transfer. The membranes were then incubated with an anti-Sh2 antibody (diluted 1:2000) and anti-actin antibody (diluted 1:10,000) for 2 h at room temperature. Subsequently, the membranes were incubated with horseradish peroxidase-conjugated anti-rabbit IgG (Kangwei Company, Chengdu, China; diluted 1:10,000) for 30 min, and the immune-reactive bands were detected using the chemiluminescent substrate Lumi-Light Immune-blotting Substrate (Thermo Scientific, Cat: PI208186 and PJ209602, Rockford, IL, USA). For the time course of the alkaline phosphatase treatment experiment, 240 µg maize endosperm total proteins (60 µg each sample) from 27 DAP and 30 DAP were treated with alkaline phosphatase indicated time. All immunoblot assays were independently performed at least three times.

### 4.6. Immunoprecipitation (IP) and Co-Immunoprecipitation (Co-IP)

Co-immunoprecipitation experiments were performed with slight modifications based on the methods outlined by [46]. Purified Sh2 antibodies, approximately 10 µg each, were utilized for immunoprecipitation and co-immunoprecipitation experiments with cell lysates obtained from 27 DAP (1 mL, 0.5 mg/mL proteins). The antibody–cell lysate mixture was incubated on a rotator at 4 °C for 4 h. To precipitate the antibodies, 20 mm of Protein A/G-Sepharose (Biorad), prepared as a 50% (*w*/*v*) slurry in phosphate-buffered saline (PBS; 137 mM NaCl, 10 mM Na_2_HPO_4_, 2.7 mM KCl, 1.8 mM KH_2_PO_4_, pH 7.4), was added and incubated at 4 °C for an additional 4 h. The Protein A/G-Sepharose/antibody/protein complex was then centrifuged at 2000× *g* for 5 min at 4 °C in a refrigerated microfuge, and the supernatant was discarded. The pellet obtained was washed five times with PBS, followed by an additional five washes with a buffer containing 10 mM HEPES-NaOH (pH 7.5) and 150 mM NaCl. The washed pellet was subsequently boiled in the SDS loading buffer, separated by SDS-PAGE, and subjected to immunoblot analysis.

### 4.7. iTRAQ^TM^ Labeling and Mass Spectrometry Analysis

Phosphorylated proteins were identified using the iTRAQ^TM^ method. The experiment was repeated with three samples of Mo17 corn 25 DAP grain protein collected during both daytime and night. Sample lysis and protein extraction were performed using an SDT buffer (4% SDS, 100 mM Tris-HCl, 1 mM DTT, pH 7.6). The protein concentration was determined using the BCA Protein Assay Kit (Bio-Rad, Hercules, CA, USA). Protein digestion was carried out using trypsin following the filter-aided sample preparation (FASP) procedure. The resulting digested peptides from each sample were desalted using C18 Cartridges (Empore^TM^ SPE Cartridges C18, standard density, ThermoFisher Scientific^TM^, Walttham MA, USA), concentrated by vacuum centrifugation, and reconstituted in 40 µL of 0.1% (*v*/*v*) formic acid. Enrichment of phosphopeptides was performed using the High-Select^TM^ Fe-NTA Phosphopeptides Enrichment Kit according to the manufacturer’s instructions (Thermo Scientific). After lyophilization, the phosphopeptides were re-suspended in 20 µL of loading buffer (0.1% formic acid). LC-MS analysis was conducted using a timsTOF Pro mass spectrometer (Bruker) coupled with Nanoelute (Bruker Daltonics, Bremen, Germany) for 60 min. The peptides were loaded onto a C18-reversed phase analytical column (25 cm long, 75 μm inner diameter, 1.9 μm, C18) with buffer A (0.1% formic acid) and separated using a linear gradient of buffer B (84% acetonitrile and 0.1% formic acid) at a flow rate of 300 g/min. The mass spectrometer operated in positive ion mode, collecting ion mobility MS spectra within the mass range of m/z 100–1700 and 1/k0 values ranging from 0.6 to 1.6. Subsequently, 10 cycles of PASEF MS/MS were performed with a target intensity of 1.5 k and a threshold of 2500. The active exclusion was enabled with a release time of 0.4 min. The original data obtained from the mass spectrometry analysis were in RAW file format, and library identification and quantitative analysis were conducted using MaxQuant software (version 1.5.3.17).

### 4.8. Yeast Two-Hybrid Assay

To design the Sh2 homologous recombination primers for yeast two-hybrid assays, we had the following sequences: F: TCTCAGAGGAGGACCTGCATATGATGCAGTTT GCACTTGCATTGGACACG, R: TGCGGCCGCTGCAGGTCGACGGATCCCTATATGA CAGACCCATCGTTGAT GGT. Similarly, for Sh2 Ser31A/D/E, the primers were as follows: F: GGCCATGGAGGCCAGTGAATTCATGGACATGGCTTT, R: TCGAGCTCG AGCTCGATGGATCCTCATATAACTGTTCCACT. To construct the yeast two-hybrid vectors, the Sh2, and Sh2 Ser31A/D/E sequences will be inserted into the pGBK-T7 vector as the bait. For Bt2, the same method was used, and the primers were: F: GGCCATGGAGGC CAGTGAATTCATGGACATGGCTTT, R: TCGAGCTCGAGCTCG ATGGATCCTCATATAACTGTTCCACT. The Bt2 sequence was inserted into the pGAD-T7 vector as the prey. The pGBK-P6 and pGBK-T7 vectors were separately transformed into yeast strains AH109. The resulting colonies were inoculated in SD/-leu-trp liquid medium. After 24 h of culture, the cells were plated on SD/-leu-trp and SD/-trp-leu-his-ade/X-α-gal plates sequentially. The colonies showing blue coloration and better growth were considered positive colonies. This indicated that SH2-Ser31A/D/E still interacts with Bt2 [47].

### 4.9. Protoplasm Preparation

Yellow-headed leaves at the seedling stage were carefully cut into leaf fillets measuring 0.5–1 mm in width using a sharp blade. For corn endosperm (7–9 DAP), the endosperm was extracted from the corn using sharp tweezers. The endosperm was then promptly divided into small pieces with a blade and placed in the enzymatic hydrolysate to prevent it from drying out. The enzymatic hydrolysate consisted of 1.5% cellulase, 0.75% pectin, 10 mM MES, 600 mM mannitol, 10 mM CaCl_2_, and 0.1% BSA, and the endosperm was enzymatically hydrolyzed for 4–6 h. After lysis, an equal volume of W5 solution (154 mM NaCl, 125 mM CaCl_2_, 5 mM KCl, 2 mM MES-KOH) was added to the leaf fillets. The mixture was then centrifuged at 4 °C and 90× *g* for 3 min, and the supernatant was discarded. Next, 5 mL of W5 solution was slowly added to re-suspend the protoplasts, followed by centrifugation at 4 °C and 90× *g* for 1 minute. After removing the supernatant, 1 mL of W5 solution was added, and the mixture was left to stand on ice for 30 min. Following this, the supernatant was removed by centrifugation, and the protoplasts were suspended in MMG solution (400 mM mannitol, 15 mM MgCl_2_, 4 mM MES-KOH, pH 5.6) at a concentration of approximately 2 × 105 protoplasts per mL of MMG solution. To initiate transformation, 10–20 μg of the plant expression vector was added to the bottom of a 2 mL round-bottom centrifuge tube, followed by the addition of 100 μL of protoplasts. The contents were gently mixed, and then 100 μL of PEG-Ca^2+^ solution (4 M mannitol, 10 mM CaCl_2_, 40% PEG) was added to the protoplasts. The induced transformation process was carried out in the dark for 15 min. Afterward, 400 μL of W5 solution was added to halt the reaction, and the mixture was centrifuged at room temperature, 90× *g* for 2 min to remove the supernatant. An additional 100 μL of W5 solution was added and gently mixed. The protoplasts were incubated in darkness at 28 °C for 12 h and observed under a microscope. Subsequently, the protoplasts were collected by centrifugation at room temperature, 90× *g* for 1 min, and a small amount was taken for fluorescence signal observation under a confocal microscope.

### 4.10. AGPase Activity Assay

To assess the activity of AGPase, the change in RLUC (Renilla luciferase) value corresponding to ATP production was measured using the ADP/ATP luminescence detection kit (Beyotime). This luminescence detection kit allows for the quantification of AGPase activity. Initially, the pUG221-Sh2 template was used in the laboratory. The following primers were employed to introduce mutations into the pUG221-Sh2 plasmid at the S31A, S31D, and S31E positions: S31A-F: AATTAGCGATTGGGGGCAGAAAGC AGG, S31A-F: CCCAATCGCTAATTTTTCCAACCTGAATCCC. S31D-F: AATTAGAT ATTGGGGGCAGAAAGCAGGAG, S31D-R: CCCCAATATCTAATTTTTCCAACCTGA ATCCC, S31E-F: AATTAGAAATTGGGGGCAGAAAGCAGGAG, S31E-R: CCCCAAT TTCTAATTTTTCCAACCTGAATCCC. Using the QuickMutation™ site-directed mutation kit (Beyotime), the pUG221-Sh2-Ser31A/D/E mutant plasmids were successfully generated. These mutant plasmids were then co-transfected into endosperm protoplasts along with pUG221-Bt2 and pUG221, respectively. After 14–16 h of expression under dark conditions, the protoplasts were collected through centrifugation at room temperature, 90× *g* for 1 minute. To determine RLUC activity, the Luciferase Assay System (Promega) was used, following the instructions provided in the ATP Assay Kit (Beyotime). GUS (β-glucuronidase) activity was determined by adding 40 μL of cell lysis supernatant to 40 μL of the reaction substrate MUG working solution. After thorough mixing, 30 μL of the mixture was taken and added to 60 μL of 0.3 M Na_2_CO_3_ solution to stop the reaction. GUS activity was measured immediately at 0 h after absorbing 75 μL of the mixture. For the remaining 50 μL of the reaction solution, the dark incubation was continued at 37 °C for 4 h. Then, 30 μL of the solution was added to 60 μL of 0.3 M Na_2_CO_3_ solution to stop the reaction, and GUS activity was measured after a total of 4 h. The terminated reaction solution was added to a black 96-well cell culture plate, and GUS activity was determined using a fluorescence spectrophotometer (Luminoskan Ascent, ThermoFisher Scientific^TM^, Walttham, MA, USA). The ratio of RLUC activity to GUS activity represents the AGPase activity.

## 5. Conclusions

In our study, we determined that the large subunit of AGPase undergoes phosphorylation specifically on the S31 site, suggesting a novel mechanism for regulating AGPase activity via Sh2 phosphorylation, and speculate that this phosphorylation could occur during the synthesis of starch in maize endosperm. Interestingly, our findings revealed that the simulated phosphorylation and dephosphorylation mutations did not influence the interaction between Sh2 and Bt2. Furthermore, concerning subcellular localization, we noted that these mutations did not impact the intracellular positioning of Sh2, which typically functions in the endosperm cytoplasm. Upon assessing AGPase activity, it was found that the activity level in simulated phosphorylation and dephosphorylation mutants was higher than in the wild type. This suggests that phosphorylation potentially enhances AGPase activity. Additionally, we observed that the simulated phosphorylation mutations of E and G exhibited superior activity compared to the simulated dephosphorylation mutation of A. These results highlight the significant role of phosphorylation in modulating the activity of AGPase, a critical enzyme in the initial step of starch synthesis. Therefore, any variations in its activity could have substantial consequences on starch production. Our study offers valuable insights that could enhance our comprehension of starch synthesis, with substantial implications for crop improvement. By generating and studying mutants with augmented AGPase activity, our research opens avenues for the development of crop varieties with potentially higher starch content. This could lead to improvements in crop yield and quality. Furthermore, elucidating how the phosphorylation status of Sh2 influences AGPase activity could pave the way for the creation of crop varieties with altered starch properties. This could potentially increase their value across various industries including food, pharmaceuticals, and biofuels.

## Figures and Tables

**Figure 1 ijms-24-15283-f001:**
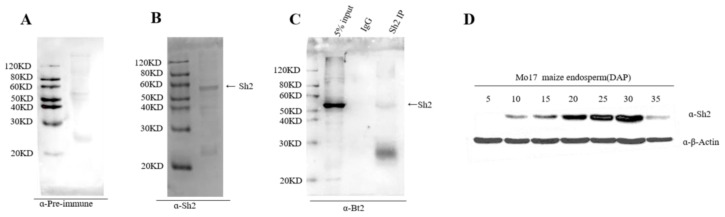
Assessment of immunoprecipitation and Sh2 expression in maize endosperm matrix protein. (**A**) Western blotting was performed on 20 DAP maize endosperm using pre-immune serum as a control. The antibody was diluted in a ratio of 1:5000; (**B**) Sh2 antibody was evaluated using Western blotting on 20 DAP maize endosperm. The antibody was diluted at a ratio of 1:5000. The arrow indicates the Sh2 protein; (**C**) Western blot of Sh2 antibody immunoprecipitation. The arrow indicates the Sh2 protein; (**D**) Expression analysis of Sh2 protein in maize Mo17 endosperm post-pollination. Sh2 and actin antibodies were diluted at ratios of 1:5000 and 1:1000, respectively.

**Figure 2 ijms-24-15283-f002:**
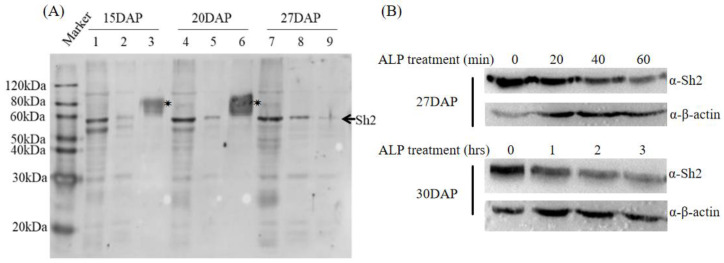
Phos-tag™-based enrichment of phosphorylated AGPase Sh2 from maize endosperm. (**A**) Sh2 antibody was used to detect the Sh2 band with a 1: 500 dilution ratio. Lanes 1, 4, and 7 represent total endosperm proteins (20 µg each) extracted from maize at 15, 20, and 27 DAP, respectively. Lanes 2, 5, and 8 depict the products of Phos-tag™ agarose enrichment from 200 µg total protein samples in maize endosperm at 15, 20, and 27 DAP, respectively. Lanes 3, 6, and 9 illustrate 200 µg maize endosperm total proteins from 15, 20, and 27 DAP, post-treatment with alkaline phosphatase and subsequent enrichment with Phos-tag™ agarose. Asterisk (*) signs are indicating the non-specific bands. (**B**). The 240 µg maize endosperm total proteins (60 µg each sample) from 27 DAP and 30 DAP were treated with alkaline phosphatase 50U indicated time. A sh2 antibody was used to detect the Sh2 band with a 1: 2000 dilution ratio and an anti-actin antibody was used to detect the actin band with a 1:10,000 dilution ratio for Western blot.

**Figure 3 ijms-24-15283-f003:**
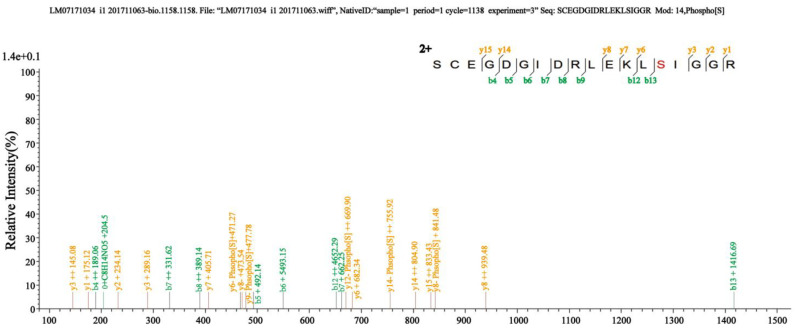
Mass spectrometry identification of Sh2 phosphopeptide. In the phosphorylated peptide “SCEGDGIDRLEKLSIGGR”, the red letter (S) is the phosphorylation site at 31.

**Figure 4 ijms-24-15283-f004:**
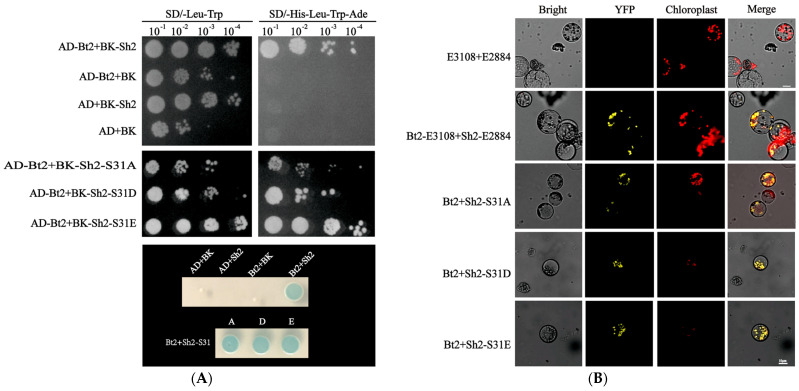
Assessing the interaction between Bt2 and Sh2-S31 mutants. (**A**) Yeast two-hybrid interaction analysis of Bt2 with Sh2-S31A/D/E mutants. Transformed yeast was plated on SD/−Leu−Trp and SD/−His−Leu−Trp−Ade gradient dilution plates (10^−1^, 10^−2^, 10^−3^, and 10^−4^), followed by X-α-Gal staining; (**B**) Investigation of the interaction and localization of Sh2 and Sh2-S31A/D/E mutants with Bt2 within maize leaf protoplasts.

**Figure 5 ijms-24-15283-f005:**
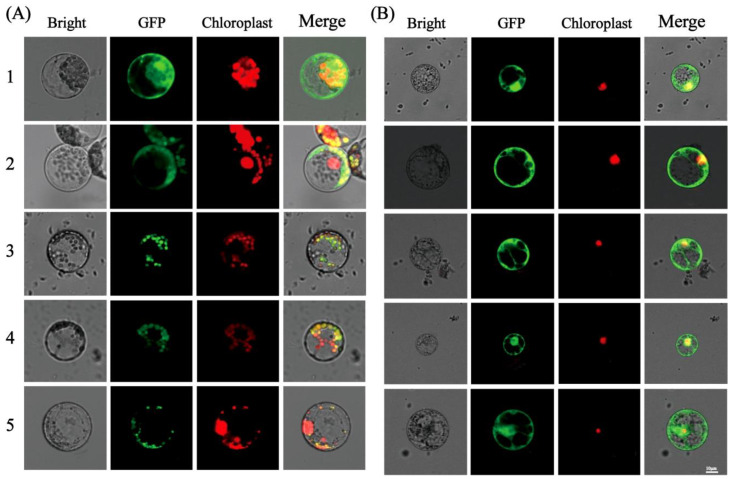
Subcellular localization analysis of Sh2 and Sh2-S31A/D/E mutants in maize endosperm protoplasts; (**A**) Immunofluorescence mapping to investigate the subcellular localization of the Sh2 and Sh2-S31A/D/E mutants in maize leaf protoplasts; (**B**) Immunofluorescence mapping to investigate the subcellular localization of the Sh2 and Sh2-S31A/D/E mutants in maize endosperm protoplasts. (1) pCAMBIA2300-35s-eGFP. (2) pCAMBIA2300-35s-Sh2-eGFP. (3) pCAMBIA2300-35s -S31A-eGFP. (4) pCAMBIA2300-35s-S31D-eGFP. (5) pCAMBIA2300-35s-S31E-eGFP.

**Figure 6 ijms-24-15283-f006:**
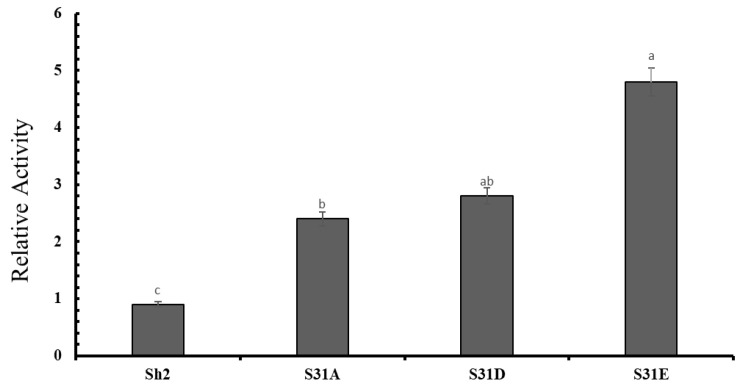
Comparison of AGPase activity between Sh2-S31A/D/E mutations and Bt2; (B) AGPase activity was measured to evaluate the impact of Sh2-S31A/D/E mutations compared to Bt2. The error bars indicate the standard error of the mean value, calculated from three independent biological replicates. Significantly different means (at *p* < 0.05) were calculated from the one-way ANOVA followed by LSD with letters indicating differences. a, ab, b and c indicate significant differences.

**Table 1 ijms-24-15283-t001:** iTRAQ^TM^ identification of AGPase large subunit Sh2 phosphorylation site.

Credibility	Peptide Sequence	Phosphorylation Site	Sh2 Site
95.0900018215179	SCEGDGIDRLEKLSIGGR	Phospho (S) @14	Serine 31

## Data Availability

Not applicable.

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
