# Peer review of "Serine 31 Phosphorylation-Driven Regulation of AGPase Activity: Potential Implications for Enhanced Starch Yields in Crops"

_ijms, 2023, doi:10.3390/ijms242015283_

Round 1

Reviewer 1 Report (New Reviewer)

In this study, the authors have determined that the large subunit of AGP-ase undergoes phosphorylation, and suppose that this phosphorylation might occur during starch synthesis in the maize endosperm. They found that the phosphorylated AGP-ase subunit possesses an affinity for Phos-TagTM agarose, while the dephosphorylated AGP-ase subunit loses its ability to bind ALP. Authors have identified the phosphorylation site, sh2-Ser31, using iTRAQTM. Regarding subcellular localisation, these mutations do not affect the intracellular positioning of Sh2, which functions in the endosperm cytoplasm. When evaluating AGP-ase activity, it was found that the level of activity in the simulated phosphorylation and dephosphorylation mutants was higher than in the wild type. This suggests that the phosphorylation potentially increases AGP-ase activity. The results highlight the significant role of phosphorylation in modulating the activity of AGP-ase, a critical enzyme in the initial step of the synthesis of starch. This study is important because it provides insights that could improve the understanding of starch synthesis, important for crop improvement. By generating and studying mutants with increased AGP-ase activity, this research can open a new way for the development of crop varieties with higher starch content. For this reason, I recommend the publication of this manuscript. However, two main types of corrections  need to be made by the authors  in this manuscript,   as follows:

1)Figure 3 is not clear. The authors must provide in their manuscript a figure with  good resolution;

2) The bibliography from the chapter entitled References is not written according to MDPI rules. Authors must respect the rules of the journal for references.

In this study, the authors have determined that the large subunit of AGP-ase undergoes phosphorylation, and suppose that this phosphorylation might occur during starch synthesis in the maize endosperm. They found that the phosphorylated AGP-ase subunit possesses an affinity for Phos-TagTM agarose, while the dephosphorylated AGP-ase subunit loses its ability to bind ALP. Authors have identified the phosphorylation site, sh2-Ser31, using iTRAQTM. Regarding subcellular localisation, these mutations do not affect the intracellular positioning of Sh2, which functions in the endosperm cytoplasm. When evaluating AGP-ase activity, it was found that the level of activity in the simulated phosphorylation and dephosphorylation mutants was higher than in the wild type. This suggests that the phosphorylation potentially increases AGP-ase activity. The results highlight the significant role of phosphorylation in modulating the activity of AGP-ase, a critical enzyme in the initial step of the synthesis of starch. This study is important because it provides insights that could improve the understanding of starch synthesis, important for crop improvement. By generating and studying mutants with increased AGP-ase activity, this research can open a new way for the development of crop varieties with higher starch content. For this reason, I recommend the publication of this manuscript. However, two main types of corrections  need to be made by the authors  in this manuscript,   as follows:

1)Figure 3 is not clear. The authors must provide in their manuscript a figure with  good resolution;

2) The bibliography from the chapter entitled References is not written according to MDPI rules. Authors must respect the rules of the journal for references.

Author Response

Reviewer 1 In this study, the authors have determined that the large subunit of AGP-ase undergoes phosphorylation, and suppose that this phosphorylation might occur during starch synthesis in the maize endosperm. They found that the phosphorylated AGP-ase subunit possesses an affinity for Phos-TagTM agarose, while the dephosphorylated AGP-ase subunit loses its ability to bind ALP. Authors have identified the phosphorylation site, sh2-Ser31, using iTRAQTM. Regarding subcellular localisation, these mutations do not affect the intracellular positioning of Sh2, which functions in the endosperm cytoplasm. When evaluating AGP-ase activity, it was found that the level of activity in the simulated phosphorylation and dephosphorylation mutants was higher than in the wild type. This suggests that the phosphorylation potentially increases AGP-ase activity. The results highlight the significant role of phosphorylation in modulating the activity of AGP-ase, a critical enzyme in the initial step of the synthesis of starch. This study is important because it provides insights that could improve the understanding of starch synthesis, important for crop improvement. By generating and studying mutants with increased AGP-ase activity, this research can open a new way for the development of crop varieties with higher starch content. For this reason, I recommend the publication of this manuscript. Author response: We are grateful to the reviewer for their careful reading of our paper and their insightful comments and suggestions. We have considered all feedback and made the necessary adjustments accordingly. For ease of identification, all changes have been written in red in the manuscript file. However, two main types of corrections  need to be made by the authors  in this manuscript,   as follows: 1)Figure 3 is not clear. The authors must provide in their manuscript a figure with  good resolution; Author response: Thank you for underlining this aspect, we have improved the figures' quality and hoping that its resolution is better now (Figure 3). Figure 3. Mass spectrometry identification of Sh2 phosphopeptide. In phosphorylated peptide “SCEGDGIDRLEKLSIGGR”, the red letter (S) is the phosphorylation site at 31. 2) The bibliography from the chapter entitled References is not written according to MDPI rules. Authors must respect the rules of the journal for references. Author response: Thank you for pointing this out. We have modified the bibliography as per journal requirements by using the endnote software. We look forward to hearing from you regarding our submission. We would be glad to respond to any further questions and comments that you may have. Thanks!

Reviewer 2 Report (New Reviewer)

Current manuscript entitled “Serine 31 Phosphorylation-Driven Regulation of AGPase Activity: Potential Implications for Enhanced Starch Yield in Crops” by “Yu et al” identified the phosphorylation site (serine at the 31st position of the linear amino acid sequence) of the AGPase large subunit (Sh2) using iTRAQTM. Subsequently, to ascertain the impact of Sh2 phosphorylation on AGPase, we carried out site-directed mutations creating Sh2-S31A (serine residue replaced with alanine) to mimic dephosphorylation and Sh2-S31D (serine residue re-placed with aspartic acid) or Sh2-S31E (serine residue replaced with glutamic acid) to mimic phosphorylation. Preliminary investigations were performed to determine Sh2 subcellular localization, its interaction with Bt2, and the resultant AGPase enzymatic activity. Our findings indicate that the localization of the Sh2-S31A and Sh2-S31D or Sh2-S31E mutations aligns with that of Sh2. Furthermore, none of these mutations at the S31 site of Sh2 seem to affect its interaction with Bt2 (smaller subunit), as demonstrated by yeast two-hybrid assays and fluorescence bimolecular complementation experiments. The manuscript seems good and can be accepted after addressing the following comments.

1.      Please remove the abbreviations in the keywords section.

2.      Clear statements of the novelty of the work should also appear briefly in the Abstract and Conclusions sections.

3.      The manuscript has grammatical errors/ typos/ incomplete sentences and non-relative phrases.

4.      The Abstract should contain answers to the following questions: What problem was studied and why is it important? What methods were used? What are the important results? What conclusions can be drawn from the results? What is the novelty of the work and where does it go beyond previous efforts in the literature? Add the main findings and objective of the current study in the abstract.

5.      Provide some more information on protein expression and purification.

6.      In between capital letters have been used especially in the section captions, please remove them.

7.      Improve the image quality of the figures.

Minor editing of English language required

Author Response

Reviewer 2

Current manuscript entitled “Serine 31 Phosphorylation-Driven Regulation of AGPase Activity: Potential Implications for Enhanced Starch Yield in Crops” by “Yu et al” identified the phosphorylation site (serine at the 31st position of the linear amino acid sequence) of the AGPase large subunit (Sh2) using iTRAQTM. Subsequently, to ascertain the impact of Sh2 phosphorylation on AGPase, we carried out site-directed mutations creating Sh2-S31A (serine residue replaced with alanine) to mimic dephosphorylation and Sh2-S31D (serine residue re-placed with aspartic acid) or Sh2-S31E (serine residue replaced with glutamic acid) to mimic phosphorylation. Preliminary investigations were performed to determine Sh2 subcellular localization, its interaction with Bt2, and the resultant AGPase enzymatic activity. Our findings indicate that the localization of the Sh2-S31A and Sh2-S31D or Sh2-S31E mutations aligns with that of Sh2. Furthermore, none of these mutations at the S31 site of Sh2 seem to affect its interaction with Bt2 (smaller subunit), as demonstrated by yeast two-hybrid assays and fluorescence bimolecular complementation experiments. The manuscript seems good and can be accepted after addressing the following comments.

Author response: We are grateful to the reviewer for their careful reading of our paper and their insightful comments and suggestions. We have considered all feedback and made the necessary adjustments accordingly. For ease of identification, all changes have been written in red in the manuscript file.

  1.      Please remove the abbreviations in the keywords section.

Author response: Thank you for pointing this out. We have modified the key words by removing the abbreviations.

Following changes have been made (Key words section).

Keywords: AGPase; shrunken2; phosphorylation; enzyme activity; subcellular localization

  1.      Clear statements of the novelty of the work should also appear briefly in the Abstract and Conclusions sections.

Author response: Thank you for pointing this out. We have made significant improvements in abstract and conclusion and hopefully both sections are incorporated with mentioned requirements.

Following changes have been made (Abstract Lines 20-25).

ADP-Glc pyrophosphorylase (AGPase), which catalyzes the transformation of ATP and glucose-1-phosphate (Glc-1-P) into adenosine diphosphate glucose (ADP-Glc), acts as a rate-limiting enzyme in crop starch biosynthesis. Prior research has hinted at the regulation of AGPase by phosphorylation in maize. However, the identification and functional implications of these sites remain to be elucidated. In this study, we identified the phosphorylation site (serine at the 31st position of the linear amino acid sequence) of the AGPase large subunit (Sh2) using iTRAQTM.

Conclusion Lines 545-561.

In our study, we determined that the large subunit of AGPase undergoes phosphorylation specifically on the S31 site suggesting a novel mechanism for regulating AGPase activity via Sh2 phosphorylation, and speculate that this phosphorylation could occur during the synthesis of starch in maize endosperm. Interestingly, our findings revealed that the simulated phosphorylation and dephosphorylation mutations did not influence the interaction between Sh2 and Bt2. Furthermore, concerning subcellular localization, we noted that these mutations did not impact the intracellular positioning of Sh2, which typically functions in the endosperm cytoplasm. Upon assessing AGPase activity, it was found that the activity level in simulated phosphorylation and dephosphorylation mutants was higher than in the wild type. This suggests that phosphorylation potentially enhances AGPase activity. Additionally, we observed that the simulated phosphorylation mutations of E and G exhibited superior activity compared to the simulated dephosphorylation mutation of A. These results highlight the significant role of phosphorylation in modulating the activity of AGPase, a critical enzyme in the initial step of starch synthesis. Therefore, any variations in its activity could have substantial consequences on starch production.

  1. The manuscript has grammatical errors/ typos/ incomplete sentences and non-relative phrases.

Author response: Thank you for pointing this out. We have made significant improvements to the language of our manuscript, and it has been thoroughly reviewed by a native English speaker. Furthermore, grammar mistakes have also been checked and corrected.

  1. The Abstract should contain answers to the following questions: What problem was studied and why is it important? What methods were used? What are the important results? What conclusions can be drawn from the results? What is the novelty of the work and where does it go beyond previous efforts in the literature? Add the main findings and objective of the current study in the abstract.

Author response: Thank you for pointing this out. We have further modified the abstract by considering most of your suggested points. Hopefully, it is more robust now.

Following changes have been made (Abstract).

Abstract: ADP-Glc pyrophosphorylase (AGPase), which catalyzes the transformation of ATP and glucose-1-phosphate (Glc-1-P) into adenosine diphosphate glucose (ADP-Glc), acts as a rate-limiting enzyme in crop starch biosynthesis. Prior research has hinted at the regulation of AGPase by phosphorylation in maize. However, the identification and functional implications of these sites remain to be elucidated. In this study, we identified the phosphorylation site (serine at the 31st position of the linear amino acid sequence) of the AGPase large subunit (Sh2) using iTRAQTM. Subsequently, to ascertain the impact of Sh2 phosphorylation on AGPase, we carried out site-directed mutations creating Sh2-S31A (serine residue replaced with alanine) to mimic dephosphorylation and Sh2-S31D (serine residue replaced with aspartic acid) or Sh2-S31E (serine residue replaced with glutamic acid) to mimic phosphorylation. Preliminary investigations were performed to determine Sh2 subcellular localization, its interaction with Bt2, and the resultant AGPase enzymatic activity. Our findings indicate that phosphorylation exerts no impact on the stability or localization of Sh2. Furthermore, none of these mutations at the S31 site of Sh2 seem to affect its interaction with Bt2 (smaller subunit). Intriguingly, all S31 mutations in Sh2 appear to enhance AGPase activity when co-transfected with Bt2, with Sh2-S31E demonstrating a substantial five-fold increase in AGPase activity compared to Sh2. These novel insights lay a foundational groundwork for targeted improvements in AGPase activity, thus potentially accelerating the production of ADP-Glc (the primary substrate for starch synthesis), promising implications for improved starch biosynthesis and holding the potential to significantly impact agricultural practices.

  1.      Provide some more information on protein expression and purification.

Author response: Thank you for pointing this out. We have added the details explaining the expression and purification of proteins.

Following changes have been made (Lines 130-153).

2.2. Protein Expression and Purification

To create the GST-gene fusion system protein expression vector PGEX-6T-1-SH2, the Sh2 gene was incorporated into the PGEX-6T-1 vector. The Sh2 homologous recombination primers used for this purpose were as follows: Sh2-F: AAAACCTGTATT TTCAGGGATCCATGCAGTTTGCACTTGCATTGGACACG, Sh2-R: CAGTCACGATG CGGCCGCTCGAGCTATATGACAGACCCATCGTTGATGGTTG. To express and purify the Sh2 protein, the procedures outlined in the GST gene fusion system handbook provided by GE Healthcare (Piscataway, NY, USA) were followed. Briefly, the E. coli BL21 transformation cells carrying the PGEX-6T-1-SH2 construct was cultured in 0.1 L of LB (Luria–Bertani) medium supplemented with 50 µg/mL of ampicillin. The culture was incubated at 37°C with continuous shaking at 150 g. Afterward, 0.5 mM IPTG was added to induce protein expression, and the incubation temperature was lowered to 28°C for 6 hours. Samples were collected every 2 hours by centrifuging bacterial cells at 4200 g for 10 minutes at 4°C. The bacterial cells were resuspended in a PBS buffer solution (137 mM NaCl, 10 mM Na2HPO4, 2.7 mM KCl, 1.8 mM KH2PO4, pH 7.4), and recombinant proteins were extracted using ultrasound. The extract was then centrifuged at 12,000 g at 4°C for 5 minutes to separate soluble proteins. Subsequently, the proteins in the total extract (before centrifugation), soluble fraction (supernatant), and insoluble fraction (precipitation) were separated using SDS-PAGE and visualized using Coomassie bright blue (CBB) staining. The purification involved separating the eluted protein using SDS-PAGE and subsequently removing and recovering the target protein bands from the gel via electro elution. Following protein concentration measurement with the Bradford reagent (catalog no. p0006, Beyotime), the recombinant protein served as an antigen for the development of a polyclonal antibody.

  1.      In between capital letters have been used especially in the section captions, please remove them.

Author response: Thank you for highlighting this aspect. We have reviewed the author guidelines for headings and captions, which state that each word in a caption should start with a capitalized letter. However, we noticed some mistakes, as some captions were in sentence case format. In the latest version, we have aligned the captions with the author guidelines. Additionally, we have thoroughly reviewed the entire manuscript and corrected these types of mistakes.

  1.      Improve the image quality of the figures.

Author response: Thank you for underlining this aspect, we have improved the figures' quality and hoping that resolution is better now.

Following changes have been made (Figures).

Figure 1.

Figure 3.

Figure 4.

Figure 5.

Comments on the Quality of English Language

Minor editing of English language required

Author response: Thank you for your thoughtful recommendations. We have made significant improvements to the language of our manuscript, and it has been thoroughly reviewed by a native English speaker. Furthermore, grammar mistakes have also been checked and corrected. We trust that the quality of the language now meets the necessary standards, and we anticipate that it will not present any further issues.

We look forward to hearing from you regarding our submission. We would be glad to respond to any further questions and comments that you may have.

This manuscript is a resubmission of an earlier submission. The following is a list of the peer review reports and author responses from that submission.

Round 1

Reviewer 1 Report

Dear Authors,

Thank you very much for the submission of the manuscript entitled "Serine 31 Phosphorylation-Driven Regulation of AGPase Activity: Potential Implications for Enhanced Starch Yield in Crops ".

I have several comments and suggestions, which could help to improve the quality of this study.

1. I kindly recommend using the reference style in accordance with the MDPI template: [1] or [2,3], or [4–6]. Moreover, please, use the MDPI format of the cited studies at the reference list.

2. P. 2, line 95

The Authors noted that "Previously in our laboratory, alkaline phosphatase treatment of samples for gel activity measurement led to lower enzymatic activity compared to the phosphorylated control, implying a relationship between AGPase activity and phosphorylation", but the data is not demonstrated. Please, give more detailed information or the relevant reference.

3. Please, use italic for in vitro and in vivo.

4. P. 3, subsection 2.3

The detailed information related to the animals should be provided, e.g. average weight, animal age, the quantity of groups and animals into each group, dosage of the antigen, etc.. In the current form above mentioned questions are not evident.

5. The resolution of Figure 5 and Figure 6 should be increased (especially for the figure inscriptions).

6. P. 12, section 4

The Discussion section is too small and could be combined with the Results section (-> Results and discussion).

Taking the above mentioned into consideration, I suggest minor revisions prior to the acceptance of this publication.

Author Response

Reviewer 1

Thank you very much for the submission of the manuscript entitled "Serine 31 Phosphorylation-Driven Regulation of AGPase Activity: Potential Implications for Enhanced Starch Yield in Crops ".

I have several comments and suggestions, which could help to improve the quality of this study.

Author response: We are grateful to the reviewer for their careful reading of our paper and their insightful comments and suggestions. We have considered all feedback and made the necessary adjustments accordingly. For ease of identification, all changes have been marked in red in the manuscript file.

Reviewer Comments to the Authors:

  1. I kindly recommend using the reference style in accordance with the MDPI template: [1] or [2,3], or [4–6]. Moreover, please, use the MDPI format of the cited studies at the reference list.

Author response: Thank you for pointing this out. We have updated the reference styles with “numbered” as per journal requirements. The changes are texted red in the manuscript file.

  1. 2, line 95

The Authors noted that "Previously in our laboratory, alkaline phosphatase treatment of samples for gel activity measurement led to lower enzymatic activity compared to the phosphorylated control, implying a relationship between AGPase activity and phosphorylation", but the data is not demonstrated. Please, give more detailed information or the relevant reference.  

Author response: Thank you for underlining this deficiency. We have updated the information and added the relevant reference Following changes have been made (Line 89-92).

Previously in our laboratory, alkaline phosphatase treatment of samples for gel activity measurement led to lower enzymatic activity compared to the phosphorylated control, implying a relationship between AGPase activity and phosphorylation [30].

[30] Yu, G., et al., The Proteomic Analysis of Maize Endosperm Protein Enriched by Phos-tagtm Reveals the Phosphorylation of Brittle-2 Subunit of ADP-Glc Pyrophosphorylase in Starch Biosynthesis Process. 2019. 20(4).

  1. Please, use italic for in vitro and in vivo.

Author response: Thank you for pointing this out. We have modified the words with italics. The changes are texted red (Lines 93, 326, 331, 342, 478, and 518).

  1. 3, subsection 2.3

The detailed information related to the animals should be provided, e.g. average weight, animal age, the quantity of groups and animals into each group, dosage of the antigen, etc.. In the current form above mentioned questions are not evident.

Author response: Thank you for underlining this deficiency. We have updated the information. Following changes have been made (Lines 137-148).

Da Shuo experimental animal company provided three months old New Zealand white rabbits (weighing 2 kg) for the study. The rabbits were housed in the Animal Core Facility, following the approved procedures of the Animal Care and Use Committee of Sichuan Agricultural University (approval no. 20160320, Chengdu, China). After one week of acclimation, rabbits were immunized subcutaneously with 500 µg of purified recombinant GST-Sh2 fusion protein emulsified with 500 µL of Freund’s complete adjuvant at a ratio of 1:1 (v/v). Two weeks after the first immunization, the rabbits were boosted with five additional subcutaneous injections with 500 µg of the purified protein mixed with 500 µL of Freund’s incomplete adjuvant per injection at a ratio of 1:1 every week. Venous blood samples were collected after three injections. The resulting antiserum containing the polyclonal maize antibody was applied to a column containing a mixture of 50% protein A and 50% protein G.

  1. The resolution of Figure 5 and Figure 6 should be increased (especially for the figure inscriptions).

Author response: Thank you for underlining this aspect, we have improved the figures' quality and hoping that its resolution is better now. Following changes have been made (Lines 409-431).

Figure 5.

Figure 6. Subcellular localization analysis of Sh2 and Sh2-S31A/D/E mutants in maize endosperm protoplasts; (A) Immunofluorescence mapping to investigate the subcellular localization of the Sh2 and Sh2-S31A/D/E mutants in maize leaf protoplasts; (B) Immunofluorescence mapping to investigate the subcellular localization of the Sh2 and Sh2-S31A/D/E mutants in maize endosperm protoplasts.  (1) pCAMBIA2300-35s-eGFP. (2) pCAMBIA2300-35s-Sh2-eGFP. (3) pCAMBIA2300-35s-S31A-eGFP. (4) pCAMBIA2300-35s-S31D-eGFP. (5) pCAMBIA2300-35s-S31E-eGFP.

  1. 12, section 4

The Discussion section is too small and could be combined with the Results section (-> Results and discussion).

Author response: Thank you for the suggestion, we have further improved the discussion section and hope that it is better now with the same headings (Lines 450-550).

We look forward to hearing from you regarding our submission. We would be glad to respond to any further questions and comments that you may have.

Thanks!

Reviewer 2 Report

Conclusions are overreaching and, in some cases, misleading. Most of the data (Figures 1, 2, 3, 4) can be just supplementary.

Lack of appropriate citations throughout the manuscript. Some examples:

Line 71: Yet, in the endosperm of grains, ADP-Glc primarily stems from cytoplasmic AGPase.

Line 80-82: Among these, Sh2 encodes the large subunit of cytoplasmic gene, 80 while AGPLE-MZM and AGPLLZM encode the large subunit in embryo and leaf tissue, 81 respectively.

Line 43: What does “environmentally friendly” and “cost-effective” starch mean?

Line 356-358: Correct the observations. Based on Fig 2D SH2 levels are increasing in endosperm up to 30 DAP and then decreasing quickly at 35 DAP. In fact, the highest protein level is observed 30 DAP.  

Lines 559-561: We are not presented with any direct evidence in this study that shows phosphorylation of AGPase regulates its activity.

Figure 1: In Fig 1B we don’t see any GST band at around 30kDa, yet we see the corresponding GST band at 0hrs TEV protease lane in Fig 1C. In fact, TEV treatment seems to have a minor effect on GST separation from SH2. How do you explain this?

Figure 2: There is no referral to figure 2c in the main text. Is this purified Bt2 protein? The figure caption says it is endosperm protein. Why and how do you do western blotting on maize endosperm Bt2 protein using the SH2 antibody? What different result do you expect to see in this experiment compared to what is presented in fig 2b? If you want to determine cross reactivity of SH2 antibody with Bt2 protein, you need to express and purify the Bt2 protein and do the western blot on this sample side by side with the purified SH2 protein.

Figure 3: I don’t see any specific increase in the signal strength of SH2 band upon Phos-tag enrichment, nor do I see any decrease in the signal strength specific to SH2 upon phosphatase treatment. All the bands on lanes 2, 5, 8 and 3, 6, 9 are weaker compared to lanes 1, 5, 7. I am not sure if lanes 2, 3, 5, 6, 8, 9 have equal amounts of protein as lanes 1, 4, and 7. The conclusions made in this section 3.3 (e.g., line 380-381) are highly misleading. In fact, the quality of Fig 3 needs to be improved. What are those smear bands on lanes 3 and 6?

Section 3.4. First paragraph: This part needs significantly more details. Please rewrite. Also, related to Table1 – Are there any other peptides that were identified as potential phosphorylation sites?

Figure 4: This can go to supplementary data.

The manuscript is written very poorly and language needs to be significantly improved.

Author Response

Comments and Suggestions for Authors

  1. Conclusions are overreaching and, in some cases, misleading. Most of the data (Figures 1, 2, 3, 4) can be just supplementary.

Author response: We appreciate the reviewer's thorough reading of our manuscript and their insightful comments and suggestions.

We have taken your feedback into account and made substantial updates to our conclusions to ensure they are more robust and directly tied to the study's findings. We have accurately revised each section of the manuscript, with the conclusion section undergoing a complete overhaul. We have diligently considered and incorporated the reviewer's suggestions and comments. For ease of reference, all changes have been highlighted in red in the manuscript file.

  1. Lack of appropriate citations throughout the manuscript. Some examples:

Line 71: Yet, in the endosperm of grains, ADP-Glc primarily stems from cytoplasmic AGPase.

Author response: Thank you for underlining this deficiency. We have included the appropriate citation to mentioned section. Moreover, various other sections are also modified and added with accurate citations. Following changes have been made (Lines 68-70).

Yet, in the endosperm of grains, ADP-Glc primarily stems from cytoplasmic AGPase [12]. Thus, AGPase is situated both in the cytoplasm and plastid of grain endosperm with a considerable proportion (85%-95%) of its activity occurring in the cytoplasm [13].

[12] Beckles, D.M., A.M. Smith, and T. ap Rees, A cytosolic ADP-glucose pyrophosphorylase is a feature of graminaceous endosperms, but not of other starch-storing organs. Plant Physiology, 2001. 125(2): p. 818-827.

[13] Sylviane, C.M. and D.J.J.o.E.B. Kay, The evolution of the starch biosynthetic pathway in cereals and other grasses. 2009(9): p. 2481-2492.

Line 80-82: Among these, Sh2 encodes the large subunit of cytoplasmic gene, 80 while AGPLE-MZM and AGPLLZM encode the large subunit in embryo and leaf tissue, 81 respectively.

Author response: Thank you for underlining this deficiency. We have included the appropriate citation to mentioned section. Following changes have been made (Lines 75-77).

Among these, Sh2 encodes the large subunit of the cytoplasmic gene, while AGPLEMZM and AGPLLZM encode the large subunit in the embryo and leaf tissue, respectively [20].

[20] Boehlein, S.K., et al., Fundamental differences in starch synthesis in the maize leaf, embryo, ovary, and endosperm. The Plant Journal, 2018. 96(3): p. 595-606.

Line 43: What does “environmentally friendly” and “cost-effective” starch mean?

Author response: Sorry this is a mistake, actually we wanted to express the nature of the starch but failed to express it appropriately. Therefore, we have updated this section and removed the confusing words. Following changes have been made (Lines 42-44).

Starch, with its abundant availability, cost-effectiveness, and ease of modification serves as a vital raw material across various sectors such as food, medicine, ethanol production, biodegradable materials, and green water treatment agents [1-3].

Line 356-358: Correct the observations. Based on Fig 2D SH2 levels are increasing in endosperm up to 30 DAP and then decreasing quickly at 35 DAP. In fact, the highest protein level is observed 30 DAP. 

Author response: Sorry, it was a mistake, and thank you for careful reading and underlining this. We have updated this section with appropriate observations. Following changes have been made (Lines 349-352).

Our Western blotting results, obtained with the Sh2 antibody, echoed this observation, revealing an increasing trend in Sh2 protein levels in the 0-30 DAP endosperm and then decreasing quickly at 35 DAP (Figure 2D). Collectively, these results validate the prepared Sh2 antibody as an effective tool for immunoblotting and IP detection.

Lines 559-561: We are not presented with any direct evidence in this study that shows phosphorylation of AGPase regulates its activity.

Author response: Thank you for underlining this. We have conducted the enzyme assay to check the effect of phosphorylation on AGPase activity by using recombinant AGPase which suggests an increase in activity (Section 3, Results; Lines 443-449). On the bases of this, we have speculated that phosphorylation can increase the AGPase activity. There were some mistakes or we failed to express our views clearly but now the complete section has been updated and we are hoping that it is more robust and well-written (Lines 525-550).

In our study, we determined that the large subunit of AGPase undergoes phosphorylation, and speculate that this phosphorylation could occur during the synthesis of starch in maize endosperm. We discovered that the phosphorylated AGPase subunit possesses a notable affinity for Phos-Tag™ agarose, whereas the dephosphorylated AGPase subunit loses its binding capacity with ALP. We identified the phosphorylation site, sh2-Ser31, with the help of iTRAQ™, suggesting a novel mechanism for regulating AGPase activity via Sh2 phosphorylation. Interestingly, our findings revealed that the simulated phosphorylation and dephosphorylation mutations did not influence the interaction between Sh2 and Bt2. Furthermore, concerning subcellular localization, we noted that these mutations did not impact the intracellular positioning of Sh2, which typically functions in the endosperm cytoplasm. Upon assessing AGPase activity, it was found that the activity level in simulated phosphorylation and dephosphorylation mutants was higher than in the wild type. This suggests that phosphorylation potentially enhances AGPase activity. Additionally, we observed that the simulated phosphorylation mutations of E and G exhibited superior activity compared to the simulated dephosphorylation mutation of A. These results highlight the significant role of phosphorylation in modulating the activity of AGPase, a critical enzyme in the initial step of starch synthesis. Therefore, any variations in its activity could have substantial consequences on starch production. Our study offers valuable insights that could enhance our comprehension of starch synthesis, with substantial implications for crop improvement. By generating and studying mutants with augmented AGPase activity, our research opens avenues for the development of crop varieties with potentially higher starch content. This could lead to improvements in crop yield and quality. Furthermore, elucidating how the phosphorylation status of Sh2 influences AGPase activity could pave the way for the creation of crop varieties with altered starch properties. This could potentially increase their value across various industries, including food, pharmaceuticals, and biofuels.

  1. Figure 1: In Fig 1B we don’t see any GST band at around 30kDa, yet we see the corresponding GST band at 0hrs TEV protease lane in Fig 1C. In fact, TEV treatment seems to have a minor effect on GST separation from SH2. How do you explain this?

Author response: Actually, the Figure 1B is the SDS-PAGE of purified GST-bound recombinant Sh2 protein and Figure 1C is the Western blot which shows a very light band of GST protein at 0h, possibly due to the ultrasonic purification of GST-bound Sh2 proteins. In many cases, ultrasonic purification can degrade the proteins so this can be a reason.

  1. Figure 2: There is no referral to Figure 2c in the main text. Is this purified Bt2 protein? The figure caption says it is endosperm protein. Why and how do you do western blotting on maize endosperm Bt2 protein using the SH2 antibody? What different result do you expect to see in this experiment compared to what is presented in fig 2b? If you want to determine cross reactivity of SH2 antibody with Bt2 protein, you need to express and purify the Bt2 protein and do the western blot on this sample side by side with the purified SH2 protein.

Author response: We are sorry as it was a huge typing mistake. We have prepared the Sh2 antibody only and done the immunoprecipitation experiment to check the affinity of the antibody to detect the Sh2 protein. It was a typo error and we have replaced the Bt2 word with Sh2 and the figure legend has also been updated. Following changes have been made (Lines 353-361).

(A)             (B)          (C)             (D)

Figure 2. Assessment of immunoprecipitation and Sh2 expression in maize endosperm matrix protein. (A) Western blotting was performed on 20 DAP maize endosperm using pre-immune serum as a control. The antibody was diluted in a ratio of 1:5000; (B) Sh2 antibody was evaluated using Western blotting on 20 DAP maize endosperm. The antibody was diluted at a ratio of 1:5000. The arrow indicates the Sh2 protein; (C) Western blot of Sh2 antibody immunoprecipitation. The arrow indicates the Sh2 protein; (D) Expression analysis of Sh2 protein in maize Mo17 endosperm post-pollination. Sh2 and actin antibodies were diluted at ratios of 1:5000 and 1:1000, respectively.

  1. Figure 3: I don’t see any specific increase in the signal strength of SH2 band upon Phos-tag enrichment, nor do I see any decrease in the signal strength specific to SH2 upon phosphatase treatment. All the bands on lanes 2, 5, 8 and 3, 6, 9 are weaker compared to lanes 1, 5, 7. I am not sure if lanes 2, 3, 5, 6, 8, 9 have equal amounts of protein as lanes 1, 4, and 7. The conclusions made in this section 3.3 (e.g., line 380-381) are highly misleading. In fact, the quality of Fig 3 needs to be improved. What are those smear bands on lanes 3 and 6?

Author response: In Figure 3 the 1, 4 and 7 lanes are actually the total protein from the endosperm that is why are more dark. However, lanes 2, 5, and 8 are the only proteins bands which are Phos-tag enrichment proteins and might because of this reason they are not very dark.

  1. Section 3.4. First paragraph: This part needs significantly more details. Please rewrite. Also, related to Table1 – Are there any other peptides that were identified as potential phosphorylation sites?

Author response: Thank you for your suggestions, we have greatly enhanced the clarity of the discussed section. Indeed, we identified four other peptides in our study, but we chose to focus on this particular one due to its higher credibility as suggested by the mass spectrum data. To provide additional support for Table 1, we have incorporated a new figure, "Figure 4", which presents the mass spectrum data for the highlighted peptide. Following changes have been made (Lines 381-389).

In a previous study, we utilized Phos-tagTM technology to enrich phosphorylated proteins from endosperm samples collected at different stages of maize pollination and identified a large number of phosphorylated proteins involved in the starch synthesis pathway through mass spectrometry analysis [30]. Among these proteins, peptides corresponding to the AGPase were detected. Several specific peptides were identified and among these “SCEGDGIDRLEKLSIGGR” was chosen for its higher credibility. Based on these findings, we speculated that Sh2 may be phosphorylated by kinases. Our results point towards the phosphorylation of Sh2 at S31 (Table 1; Figure 4). This finding elucidates that the phosphorylation of Sh2 which may occur at S31 in the maize endosperm.

Figure 4. Mass spectrometry identification of Sh2 phosphopeptide. In the phosphorylated peptide “SCEGDGIDRLEKLSIGGR”, the red letter (S) is the phosphorylation site at 31.

  1. Figure 4: This can go to supplementary data.

Author response: Thank you for the suggestion. We have moved this Figure to supplementary data. Following changes have been made (Supplementary data; Figure S1).

Figure S1. Sequencing results for Sh2 site mutations integrated into the puG-221 vector. (1) Represents the reference Sh2 sequence from the NCBI database. (2) Highlights the sequence region encompassing the Ser31 site. (3) Demonstrates the primers used for site-directed mutagenesis. (4) Provides the sequencing outcomes confirming successful mutations.

Comments on the Quality of English Language

The manuscript is written very poorly and language needs to be significantly improved.

Author response: Thank you for your thoughtful recommendations. We have made significant improvements to the language of our manuscript, and it has been thoroughly reviewed by a native English speaker. Furthermore, grammar mistakes have also been checked and corrected. We trust that the quality of the language now meets the necessary standards, and we anticipate that it will not present any further issues.

We look forward to hearing from you regarding our submission. We would be glad to respond to any further questions and comments that you may have.

Thanks!

Round 2

Reviewer 2 Report

Figure 3 (and to some extent Figure 7) is the most important data in this manuscript. However, my concerns and questions regarding Figure 3 were not fully addressed. All the lanes (1-9) in Fig 3 must be loaded with equal amounts of protein, whether treated or non-treated, to reach the conclusions in section 3.3. We are given some information at lines 190-193 and 382-385 regarding this, but what is the final amount of protein loaded in lanes 2, 3, 5, 6, 8, and 9?

Also, the quality of this figure must be improved. The authors should address the smear bands in lanes 3 and 6.

Figures 1 and 2 are just supplementary material that only shows antibody is working.

Regarding Figure 6: Why would you think that a single mutation could change the subcellular localization of the Sh2 protein when, presumably,  it already lacks a transit peptide since it is the cytosolic isoform?

The jargon used in the manuscript still fails to meet the standards of scientific language, and therefore the writing still needs improvement. Examples:

Lines 391-392

Several specific peptides were identified and among these “SCEGDGIDRLEKLSIGGR” was chosen for its higher credibility.

Line 43

Starch, with its abundant availability, cost-effectiveness,...

Author Response

Comments and Suggestions for Authors

Author response: We appreciate the reviewer's thorough reading of our manuscript again and their insightful comments and suggestions. 

Figure 3 (and to some extent Figure 7) is the most important data in this manuscript. However, my concerns and questions regarding Figure 3 were not fully addressed. All the lanes (1-9) in Fig 3 must be loaded with equal amounts of protein, whether treated or non-treated, to reach the conclusions in section 3.3. We are given some information at lines 190-193 and 382-385 regarding this, but what is the final amount of protein loaded in lanes 2, 3, 5, 6, 8, and 9?

Author response: Thank you for your meticulous review and insightful comments on our manuscript. We would like to clarify the protein loading in our experiments:

For lanes 1, 4, and 7, we loaded a consistent amount of 20μg total protein in each, serving as our positive controls. In contrast, for lanes 2, 3, 5, 6, 8, and 9, we initially incubated 200μg of total protein with Phos-tag™ beads to enrich phosphorylated proteins including phosphorylated Sh2. when total protein treated with Alkaline phosphatase, phosphorylation of protein will be removed phosphate group to dephosphorylation of protein. Phos-tag™ beads  can not enrich unphoshorylation of protein. So, for 3,6 and 9 lane, it should be negative control for dephosphorylation of protein.  It's important to note that our objective was qualitative detection of protein phosphorylation rather than a quantitative assessment of proteins. As such, variations in protein loading can be observed. We can only roughly estimate the level of phosphorylated proteins.

To provide a clearer understanding, here's a detailed description of the experimental procedure we followed:

We hope this addresses your concerns and provides clarity on our methodology.

Lines 183-192.

To enrich phosphoproteins, Zn2+-Phos-tagTM agarose was utilized following the in-structions provided by the manufacturer, Wako Pure Chemical Industries Ltd., Hiro-shima, Japan. In brief, a total maize endosperm cell lysate sample containing 200µg of protein was mixed with 200mm of Zn2+-Phos-tagTM agarose. For the dephosphorylation assay, alkaline phosphatase was added to the cell lysate. The binding assay was incubated at 4°C for 4 hours, followed by three washes with a washing buffer (0.1M Tris-CH3COOH, 1.0M CH3COONa, pH 7.5). Elution buffer (0.1M Tris-CH3COOH, 1.0M NaCl, 10mM NaH2PO4-NaOH, pH 7.5) was utilized to elute the phosphoproteins. The proteins bound to Zn2+ Phos-tagTM agarose were separated by electrophoresis. Additionally, 20µg of maize endosperm cell lysates were subjected to SDS-PAGE as a control.

Also, the quality of this figure must be improved. The authors should address the smear bands in lanes 3 and 6.

Author response: Thank you for underlining this deficiency again. We have tried to improve the quality.  In actually, we repeat this experiment for many times,  this figure is the best. Previously, we forgot to demonstrate about smear bands. We have demonstrated those bands as non-specific bands as we were unable to identify these bands. Now, we just look as non-specific bands. Following changes have been made (Section Results; Lines 378-385).

Figure 2. Phos-tag™-based enrichment of phosphorylated AGPase Sh2 from maize endosperm. Sh2 antibody was used to detect the Sh2 band with a 1: 500 dilution ratio.  Lanes 1, 4, and 7 represent total endosperm proteins (20µg each) extracted from maize at 15, 20, and 27 DAP, respectively. Lanes 2, 5, and 8 depict the products of Phos-tag™ agarose enrichment from 200µg total protein samples in maize endosperm at 15, 20, and 27 DAP, respectively. Lanes 3, 6, and 9 illustrate 200µg maize endosperm total proteins from 15, 20, and 27 DAP, post-treatment with alkaline phosphatase and subsequent enrichment with Phos-tag™ agarose. Asterisk (*) signs are indicating the non-specific bands.

Figures 1 and 2 are just supplementary material that only shows antibody is working.

Author response: Thank you for the suggestion again. We have shifted the figure 1 to the supplementary materials as Figure S1. Figure 2 is still in the manuscript as it also showing the levels of protein expression in different developmental stages.

Regarding Figure 6: Why would you think that a single mutation could change the subcellular localization of the Sh2 protein when, presumably, it already lacks a transit peptide since it is the cytosolic isoform?

Author response: Thank you for quarry. We think, a single site mutation can potentially change the subcellular localization of a protein, even if it is a cytosolic isoform that presumably lacks a transit peptide. There could be several reasons:

A mutation might introduce a new sequence that acts as a localization signal, directing the protein to a different subcellular compartment.

The mutation might disrupt interactions with other proteins that keep the protein in the cytosol. Without these interactions, the protein might be free to move to other cellular compartments.

Some mutations can create or abolish sites for post-translational modifications like phosphorylation or ubiquitination. These modifications can influence protein localization.

Mutations can affect protein stability. A less stable protein might be more rapidly degraded in the cytosol, or it might be sequestered to specific cellular compartments for degradation.

The mutation might induce a conformational change that exposes or hides certain motifs or domains, influencing interactions and localization.

The mutation might alter the binding affinity of the protein for its target proteins or molecules, which could influence its localization.

It is worth noting that while transit peptides are common signals for directing proteins to specific organelles (like chloroplasts), they are not the only determinants of protein localization. Many proteins are localized to specific cellular compartments through other sequence motifs or through interactions with other proteins, that is why we have conducted this experiment.

Chaffey, Nigel. "Alberts, B., Johnson, A., Lewis, J., Raff, M., Roberts, K. and Walter, P. Molecular biology of the cell. 4th edn." (2003): 401-401.

Scott, Michelle S., Peter V. Troshin, and Geoffrey J. Barton. "NoD: a Nucleolar localization sequence detector for eukaryotic and viral proteins." BMC bioinformatics 12.1 (2011): 1-7.

Comments on the Quality of English Language

The jargon used in the manuscript still fails to meet the standards of scientific language, and therefore the writing still needs improvement.

Author response: Thank you for underlining this deficiency again. We have made significant improvements to the language of our manuscript, and it has been thoroughly reviewed by a native English speaker again. Furthermore, grammar mistakes have also been checked and corrected again. We trust that the quality of the language now meets the necessary standards, and we anticipate that it will not present any further issues.

Examples:

Lines 391-392

Several specific peptides were identified and among these “SCEGDGIDRLEKLSIGGR” was chosen for its higher credibility.

Author response: Thank you for underlining this, we have updated the information. Following changes have been made (Lines 380-382).

During mass spectrometry analysis, numerous peptides were identified. Among them, a particular peptide that contains S31 was chosen for further investigation due to its increased likelihood of phosphorylation.

Line 43

Starch, with its abundant availability, cost-effectiveness,

Author response: Sorry again as we failed to express it appropriately. We have updated the complete sentence. Following changes have been made (Lines 43-45).

Starch, owing to its abundant availability and ease of modification, plays a crucial role as a primary raw material across multiple sectors, including food, medicine, ethanol production, biodegradable materials, and green water treatment agents [1-3].

Further, we have updated the manuscript with appropriate citations in the text. We look forward to hearing from you regarding our submission. We would be glad to respond to any further questions and comments that you may have.

Thanks!

Round 3

Reviewer 2 Report

My concern regarding Figure 2 still remains. All the lanes (1-9) in Fig 3 must be loaded with equal amounts of protein, whether treated or non-treated, to reach the conclusions in section 3.3. Therefore, this experiment must be repeated.

N/A